# Hormonal steroids induce multidrug resistance and stress response genes in *Neisseria gonorrhoeae* by binding to MtrR

Grace M. Hooks [1], Julio C. Ayala[2,7], Concerta L. Holley[2], Vijaya Dhulipala[2], Grace A. Beggs [3], John R. Perfect[4], Maria A. Schumacher [1], William M. Shafer[2,5,6] & Richard G. Brennan [1]

Transcriptional regulator MtrR inhibits the expression of the multidrug efflux pump operon *mtrCDE* in the pathogenic bacterium *Neisseria gonorrhoeae*. Here, we show that MtrR binds the hormonal steroids progesterone, β-estradiol, and testosterone, which are present at urogenital infection sites, as well as ethinyl estrogen, a component of some hormonal contraceptives. Steroid binding leads to the decreased affinity of MtrR for cognate DNA, increased *mtrCDE* expression, and enhanced antimicrobial resistance. Furthermore, we solve crystal structures of MtrR bound to each steroid, thus revealing their binding mechanisms and the conformational changes that induce MtrR.

The pathogen *Neisseria gonorrhoeae* is a Gram-negative diplococcus that colonizes genital, rectal, and oral mucosa[1]. As a strict human pathogen, *N. gonorrhoeae* is finely adapted for survival in its sole natural host and infects over 85 million individuals (both sexes) each year worldwide[2] and infections in the US are far more prevalent in women than men[3]. No vaccine is available, and reinfection by identical gonococcal strains is possible[1,4]. After successful colonization by gonococci (GC), a female patient may experience complications arising from infection including pelvic inflammatory disease, ectopic pregnancy, and infertility[1]. Neonatal blindness caused by maternal transmission during childbirth can also occur without appropriate preventative measures[1].

Since the late 1930's with the availability of sulfonamides followed by penicillin, antibiotics have been used to treat gonococcal infections, but their effectiveness has decreased with the rise of multidrug resistance (MDR)[5]. To date, *N. gonorrhoeae* has developed resistance to sulfonamides, macrolides, aminoglycosides, beta-lactams, tetracyclines and fluoroquinolones[1,6]. Currently, the sole empiric treatment for gonococcal infections in the US is ceftriaxone but alarmingly, ceftriaxone-resistant strains have been identified in Denmark, France,

Japan, Thailand, and the United Kingdom[7–12]. This consistent rise in MDR indicates the possibility of untreatable strains of *N. gonorrhoeae* and has prompted the CDC (Centers for Disease Control and Prevention) to flag this bacterium as an urgent public health threat. Thus, a detailed molecular understanding of gonococcal MDR mechanisms and their regulation are crucial to inform public health decisions[7].

*N. gonorrhoeae*, like most pathogens, contains a number of mechanisms that lead to antibiotic resistance including β-lactam antibiotic resistance by antibiotic modifying proteins[13], ribosome protective proteins[14], mutations in penicillin binding proteins[6], and reduced permeability of cell walls by outer membrane proteins[6,15]. Significantly, the direct export of antibiotics and toxic molecules by multidrug efflux pumps is a major contributor to gonococcal survival and MDR. Multidrug efflux pumps decrease the cytosolic and periplasmic concentration of antibiotics and cytotoxins by pumping them into the extracellular milieu[16]. These systems can recognize and efflux a wide variety of chemically and structurally dissimilar molecules[16]. The best characterized multidrug efflux system of *N. gonorrhoeae* is the MtrCDE tripartite system, a member of the Resistance-Nodulation-Division (RND) family[1]. Studies investigating MDR in *N. gonorrhoeae*

[1]Department of Biochemistry, Duke University School of Medicine, Durham, NC, USA. [2]Department of Microbiology and Immunology, Emory University School of Medicine, Atlanta, GA, USA. [3]Department of Molecular Biology, Princeton University, Princeton, NJ, USA. [4]Division of Infectious Diseases, Department of Medicine, Duke University Medical Center, Durham, NC, USA. [5]Laboratories of Microbial Pathogenesis, VA Medical Research Service, Veterans Affairs Medical Center, Decatur, GA, USA. [6]Emory Antibiotic Resistance Center, Emory University School of Medicine, Atlanta, GA, USA. [7]Present address: STD Laboratory Reference and Research Branch, Division of STD Prevention, Centers for Disease Control and Prevention, Atlanta, GA, USA. ✉e-mail: richard.brennan@duke.edu

report significant increases in gonococcal susceptibility to antibiotics when the MtrCDE system is impaired or deleted[17]. Overexpression of the MtrCDE efflux system due to *cis*- or *trans*-acting regulatory systems is a leading cause of gonococcal resistance to hydrophobic agents (HAs), fatty acids, and even human antimicrobial peptides such as LL-37[17–19]. However, the energetic cost of synthesizing efflux pumps is significant, and their expression is regulated tightly leading to optimal cell viability[20].

The *mtrCDE* operon is regulated by the multiple transferable resistance repressor, MtrR, which also acts as a global transcriptional regulator in *N. gonorrhoeae*[17]. Beyond its regulation of the expression of the *mtrCDE* operon, MtrR directly represses *rpoH*, which encodes a secondary sigma factor that controls the stress response in gonococci[21–23]. MtrR can regulate these genes by binding a degenerate consensus sequence upstream of their coding regions thus blocking access of RNA polymerase to the promotor[17,21]. Upon binding to cytotoxic molecules and likely sensing oxidative stress, MtrR undergoes a conformational change to relieve repression of its target genes (Fig. 1A)[18].

A member of the ubiquitous TetR family, MtrR is entirely alpha helical and comprises a helix-turn-helix (HTH) DNA binding motif and a C-terminal ligand binding/dimerization domain (Fig. 1B, C)[18]. Mutations in the *mtrR* gene or its cognate promoters are present in many clinical isolates resistant to antimicrobials and antibiotics[24]. Recent structural and biochemical studies on the MtrR-*rpoH* and MtrR-*mtrCDE* operator complexes reveal the basis of loss of function mutations in the HTH DNA binding motif that abolish regulation of these genes by MtrR[25].

Despite these initial studies, little is known about the induction and ligand recognition mechanisms of MtrR by innate human inducers. Previous work from our laboratories has shown that MtrR binds and is induced by the bile salts chenodeoxycholate (CDCA) and taurodeoxycholate (TDCA), found at extra-urogenital GC infection sites, but not glycocholate[18]. However, physiologically relevant inducers of MtrR that are found at urogenital sites, which form the vast majority of the sites of gonococcal infection, are not known, posing a significant gap in our understanding of how MtrR contributes to MDR and gonococcal survival in the reproductive tract. To address this critical lack of knowledge, we carried out a series of structural, biochemical, and cellular experiments to elucidate the binding and induction mechanisms of MtrR by female and male steroidal hormones as they are physiologically relevant molecules that would confront gonococci during infection. In the female reproductive tract, gonadol hormones are responsible for the coordination of immune cell function, vaginal microbiome, and epithelial cell architecture during menstration to optimize maternal protection against pathogen infection and also fetal implantation and survival[26–28]. This results in a "window of vulnerability" where the immune protection is dampened in the reproductive tract in order to optimize sperm and fetal survival, which can be taken advantage of by pathogens[26,27]. Thus, pathogens can utilize hormones to regulate gene expression involved in cytotoxic stress defenses allowing host infection[26,29–34]. Mechanisms for sensing and coping with cytotoxic stresses are crucial for survival of all organisms, particularly obligate human pathogens like *N. gonorrhoeae* that are routinely exposed to cytotoxic attacks by the host and inhabit an environment that is well equipped with antimicrobial defense mechanisms. Elucidation of the functions of MtrR will contribute to the understanding of MDR in *N. gonorrhoeae*, and transcriptional regulation that promotes pathogen survival. Here, we show that binding of human steroidal hormones results in a comformational change in the MtrR structure rendering it incompatible with DNA binding, thus revealing the structural mechanism of MtrR induction.

## Results

### Steroidal hormones bind MtrR and reduce its DNA-binding affinity

We first investigated the ability of MtrR to bind selected ligands. Because bacterial regulatory proteins of efflux systems are commonly induced by the substrates of the corresponding pump, allowing increased expression and resistance under cytotoxic stresses, we hypothesized that some ligands of MtrCDE are MtrR inducers[35–37]. Previous work has shown that mutations of the *mtrD* gene result in a significant decrease in the minimal inhibitory concentration (MIC) of progesterone (PTR) in *N. gonorrhoeae*[38]. Other physiologically relevant candidates of MtrR inducers include the gonadal steroids β-estradiol (EST) and testosterone (TES), which, in addition to PTR, have been shown to arrest *N. gonorrhoeae* growth[39]. Of these three, PTR is the most effective in growth inhibition, highlighting this compound as a potent protective agent against gonococcal infection. The effect of PTR may also explain the higher occurrence of asymptomatic gonorrheal infection in women than men[39]. Mutations in the *mtrR* gene also confer increased susceptibility to PTR in plated cells as well as decreased survival in the lower genital tract of female mice[40]. Additionally, female mice models are most susceptible to gonococcal infection during periods of the menses cycle when EST levels are highest[41], and mice treated with EST showed significantly increased susceptibility to disseminated gonococcal infection[42].

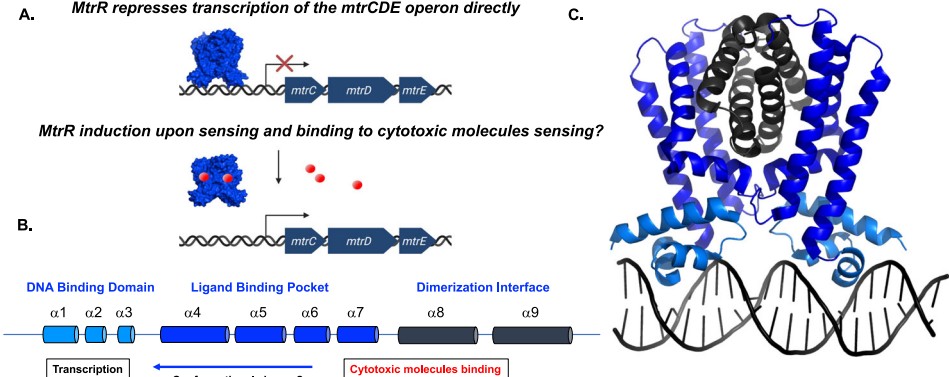

**Fig. 1 | Proposed mechanism of the transcriptional regulation of the *mtrCDE* multidrug efflux pump genes by MtrR. A** MtrR bound to the *mtrCDE* operator site blocks recognition of the −35 and −10 elements by RNA polymerase thereby precluding transcription of the *mtrC*, *mtrD*, and *mtrE* genes, which encode the periplasmic accessory, efflux, and outer membrane channel proteins, respectively, of the MtrCDE efflux system. The dark blue arrows depict the coding regions of each gene. The transcription start site is represented by a bent arrow. Transcriptional repression is shown as a red X, cytotoxic molecules are represented as red spheres, and MtrR is represented as a blue surface. **B** The entirely helical structure of MtrR is shown as cylinders and the biochemical and biological functions of each helix is described. **C** Structure of the homodimeric MtrR bound to the *mtrCDE* operator site with the helices colored as in **B** (PDB: 7JU3)[25].

We conducted isothermal titration calorimetry (ITC) studies on EST, PTR, and TES to determine their affinities for MtrR. These studies revealed that MtrR bound these steroids with equilibrium dissociation constants ($K_d$) of $2.75 \pm 0.7\,\mu M$ (PTR), $1.67 \pm 1.1\,\mu M$ (EST), and $2.26 \pm 0.3\,\mu M$ (TES) (Fig. 2A), whilst the chemical precursor cholesterol, the stress hormone cortisol, and the antibiotic azithromycin (AMR), show no binding (Fig S1). The stoichiometry for EST and TES is one molecule of steroid per MtrR monomer and intriguingly one molecule of PTR per MtrR dimer. In addition to these three gonadal steroids, we also tested the binding of ethinyl estradiol (NDR), which is a synthetic estrogen used in some hormonal contraceptives[43], and found that it binds MtrR with a $K_d$ of $0.94 \pm 0.6\,\mu M$ (Fig. 2A). These results show the substantial affinity and specificity of MtrR for steroidal hormones. To determine if these identified host ligands result in the biochemical induction of MtrR, we measured the binding affinity of MtrR for the *rpoH* and *mtrCDE* operators in the presence and absence of $125\,\mu M$ PTR, EST, TES, or NDR (in 1% methanol (MeOH)) using a DNA binding fluorescence polarization-based assay (FP). Results from these FP experiments showed up to a 13-fold and 3-fold decrease in binding affinity of MtrR for the *rpoH* and *mtrCDE* operator, respectively, in the presence of each steroid compared to binding of MtrR to these sites in the presence of 1% MeOH alone (Fig. 2B, S2). At higher concentrations, MtrR begins to bind nonspecifically to the DNA, possibly explaining why MtrR shows a more modest fold change in binding affinity for the *mtrCDE* operator compared to *rpoH*. These data support the hypothesis that gonadal steroids are physiologically relevant inducers of MtrR regulated genes.

## Structures of MtrR-steroid complexes

We carried out crystallographic studies to determine the binding mechanisms and structural basis of induction of MtrR bound to progesterone, testosterone, β-estradiol, and ethinyl estrogen. The structures of MtrR bound to cognate DNA and an "apo" form have been previously solved in our laboratory[18,25]. The structure of MtrR bound to EST was determined at a resolution of 2.37 Å using de novo phasing by single wavelength anomalous dispersion after selenomethionine substitution of the protein. Structures of MtrR bound to PTR, TES, and NDR were subsequently solved by molecular replacement at resolutions of 2.31, 2.22, and 3.20 Å, respectively. The $R_{work}$ and $R_{free}$ values for each structure are 19.2% and 24.4% (MtrR+EST), 20.8% and 26.1% (MtrR+TES), 21.0% and 25.1% (MtrR+PTR), and 20.1% and 27.7% (MtrR+NDR), respectively. The asymmetric unit (ASU) of all crystals contained two homodimers. The root mean square deviation (RMSD) of the pairwise alignments of all corresponding Cα atoms of the dimers within each ASU is 0.29 Å, 0.30 Å, 0.31 Å, and 0.48 Å for the MtrR-EST, MtrR-TES, MtrR-PTR and MtrR-NDR structures, respectively.

MtrR is composed of 210 residues and exists as a biochemical dimer of 48 kDa as confirmed by gel filtration chromatography[18,25]. Consistent with the apo structure of MtrR, the first 8 to 10 residues and a stretch of 5 to 11 residues between helices α4 and α5 are not observed in the crystal structures[18]. In all structures, the last observed carboxy-terminal residue is 209 or 210. In agreement with previous structures and other TetR family members, these MtrR-ligand structures are comprised of homodimers, composed of nine α-helices. The residues of each α-helix are: α1: 10–26, α2: 33–39, α3: 44–50, α4: 53–78, α5: 84–102, α6: 103–111, α7: 123–148, α8: 158–180, α9: 185–203 (Fig. 3A). The first three helices form the DNA binding domain which contains a HTH motif (Fig. 3A). Helices α8-α9 form the majority of the dimerization interface[18,25]. The "induced" conformation of MtrR bound to each steroid is structurally identical as the RMSDs of the alignment of all Cα atoms of each dimer in all steroid-bound structures reveals essentially the same structure: MtrR+EST:MtrR+TES = 0.43 Å, MtrR+EST:MtrR+PTR = 0.22 Å, MtrR+TES:MtrR+PTR = 0.37 Å, MtrR+EST:MtrR+NDR = 0.46 Å, MtrR+NDR:MtrR+TES = 0.50 Å, and MtrR+NDR:MtrR+PTR = 0.47 Å (Fig. 3A).

The MtrR-steroid crystal structures reveal that MtrR binds one ligand per protomer as observed in many TetR family members, including the founding member TetR, which binds one molecule of $Mg^{2+}$-tetracycline per subunit[36]. However, not all members of the TetR family bind one molecule per protomer including QacR, which notably binds one molecule per dimer[35]. The stoichiometry of MtrR bound to EST, TES and NDR is 1 ligand:1 protomer. By contrast, only one PTR is unambiguously assigned to bind the MtrR dimer. Inspection of the binding pocket of the non PTR bound-protomers reveals poor density that is possibly CAPS, a component of the crystallization buffer or a second weakly bound PTR. Interestingly, the second dimer in the asymmetric unit contains a progesterone molecule in each binding pocket. The stoichiometry for this MtrR-PTR complex is inconsistent with the ITC experiments ($n = 0.5$) and the structures ($n = 1$). This could be in part because MtrR is crystallized in an excessive concentration of steroids which does not reflect its more physiological binding, the apparent progesterone concentration in the ITC experiments is not correct, or that we are observing partial occupancy of each binding site in the crystal structure. Regardless, the MtrR-PTR complex assumes an induced conformation.

It should also be noted that the electron density for PTR was the weakest of the steroidal hormones, possibly indicating more than one binding pose for this steroid. Furthermore, β-estradiol and testosterone are positioned with the five-membered ring pointing towards the dimerization surface whilst progesterone is positioned with the five-membered ring positioned towards the lateral side of the protein (Fig. 3C, D). This indicates a binding pocket that is malleable, allowing different ligands of the same class to bind in multiple orientations, a finding which is consistent with other TetR multidrug binding proteins such as QacR of *Staphylococcus aureus* and RamR of *Salmonella enterica*[35,44].

The ligand binding pockets of each protein-steroid complex are similar in size with volumes of ~1040 Å³ (EST), ~1130 Å³ (TES), ~1000 Å³ (PTR), and ~900 Å³ (NDR) and consist of a tunnel between helices 4 and 7 that leads to a cavity surrounded by helices 4–9 (Fig S3)[45]. The ligand binding pocket is mainly comprised of hydrophobic residues with several amino acid residues in proximity to the steroids to provide Van der Waals contacts. Notably, the aromatic side chain of W136 stacks over the rings of each steroid (Fig. 3C). Interestingly, residue W136 in the MtrR-TES structure is found in two conformations within different binding pockets in the ASU, but maintains Van der Waals contacts to the steroid in each pose. Analysis of the composite omit maps of the MtrR-TES crystal structure reveal that these conformations are distributed equally in the asymmetric unit indicating that there is no significant preference for either indole ring conformation. No positively charged side chains are found in the core of the binding pocket, however, three residues, K167, R176′ (′ indicating the residue is from the other protomer of the dimer), and R137 coordinate a phosphate ion outside the ligand binding pocket in all subunits, although the density is ambiguous in the MtrR-NDR structure, likely due to the limited resolution of the structure, and was thus omitted in the refinement step for this structure. W136 and R176 were previously shown to be important in binding the bile salt chenodeoxycholate[18]. One negatively charged amino acid residue, D171 on helix α8 near the dimerization surface is positioned ~3.4 Å from the hydroxyl group on the five-membered ring of testosterone and estradiol and ~3.8 Å away from the carbonyl on C3 of progesterone. The proximity of the side chain of D171 and the hydroxyl groups of estradiol and testosterone, indicated a role for this MtrR residue in ligand specificity. In addition to D171, the side chain of glutamine residue, Q133, adopts multiple conformations that allow it to engage in hydrogen bonds to the carbonyl 1 of progesterone and the hydroxyl group of testosterone (Fig. 3C). As noted, the ligand binding pocket is predominantly formed by nonpolar and aromatic residues that engage in hydrophobic contacts with

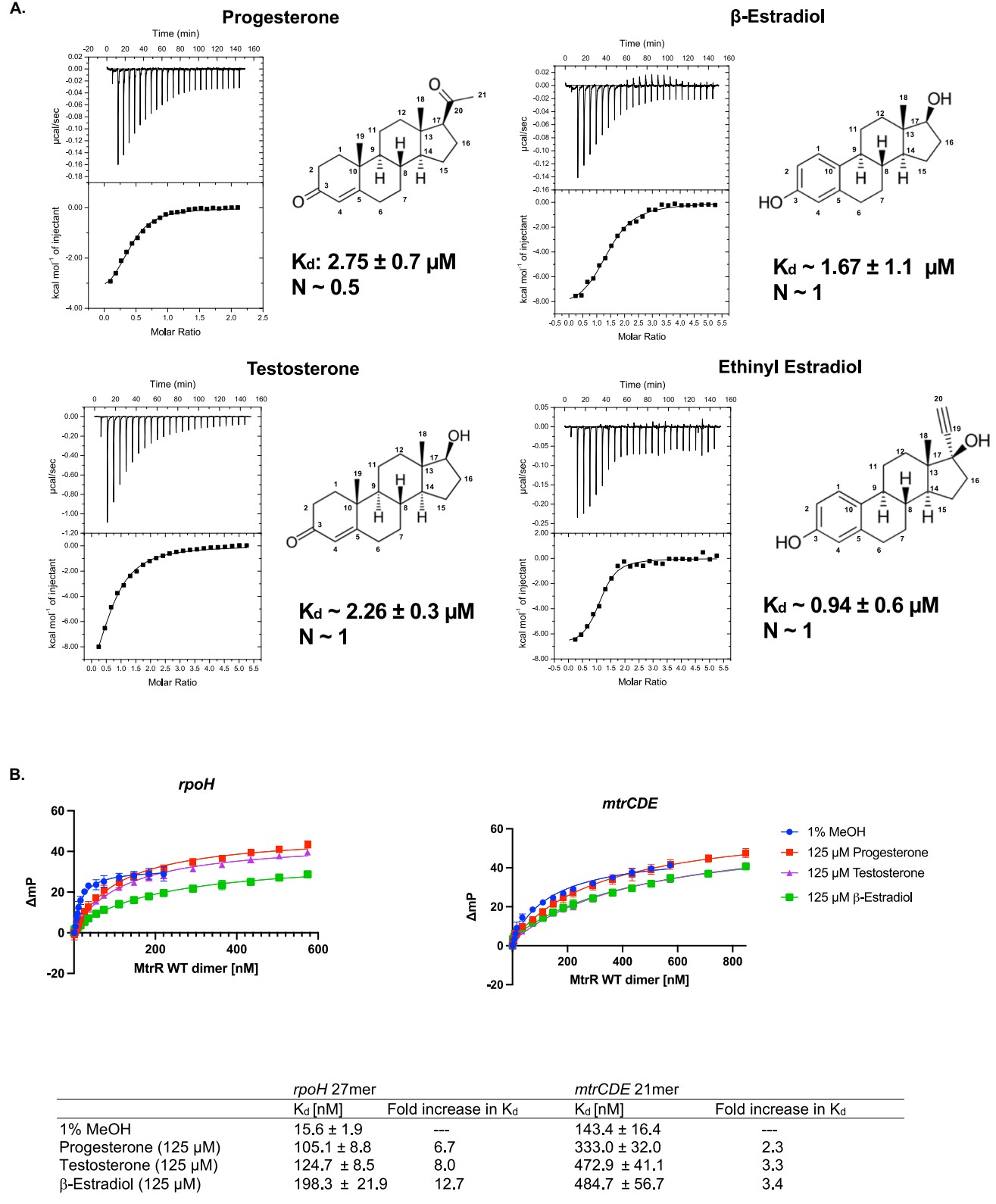

**Fig. 2 | MtrR binding to the steroids Progesterone, β-Estradiol, Testosterone, and Ethinyl Estrogen. A** Representative Isothermal titration calorimetry thermograms and binding isotherms of each natural steroid and ethinyl estradiol. At least three experimental measurements, technical and some biological replicates, were averaged for each reported binding constant. **B** Fluorescence polarization-based DNA-binding isotherms reveal significant decreases in the binding affinity of MtrR for the *rpoH* and *mtrCDE* operators in the presence of each gonadal steroid. Data are represented by the mean values (point) +/- SEM (error bar) of at least three independent experimental measurements and were averaged for each reported binding constant.

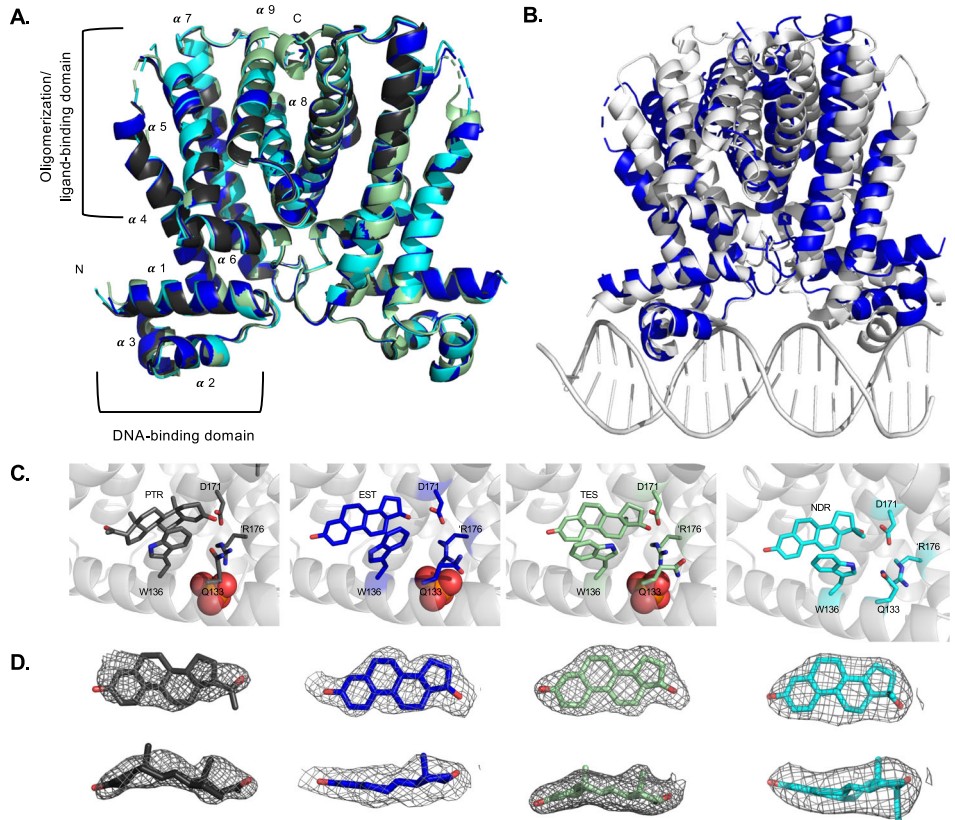

**Fig. 3 | MtrR binding and induction by gonadal steroids and ethinyl estrogen.**
**A** Overlay of hormonal steroid-bound MtrR structures with progesterone-colored charcoal, β-estradiol blue, testosterone pale green, and ethinyl estrogen cyan. The helices of one subunit are labeled **B** Overlay of MtrR bound to β-estradiol, the induced conformation, (blue), and MtrR bound to the *mtrCDE* operator site, the repressor conformation, (white). **C** View of MtrR binding to progesterone,

β-estradiol, testosterone, and ethinyl estradiol (PTR, EST, TES, and NDR). Selected binding residues are displayed as sticks. A nearby phosphate ion is shown as a sphere. **D** Electron density maps of each ligand in the MtrR ligand binding pocket looking both parallel and perpendicular to the four-membered ring of each steroid. Oxygen atoms are colored red. The composite 2Fo-Fc omit map contoured to 2.0 σ is shown in gray mesh.

the ligands. Residues that are involved in contact with all four steroids are F95, W136, I174, T91, I140, L170, and L92 (Fig S4).

Superposition of the structures of steroid-bound MtrR and MtrR bound to the *rpoH* operator sequence reveals a significant conformational change in the DNA binding domain (Fig. 3B). The center-to-center distance between the "recognition" helices of the HTH motifs in the DNA bound structure is 36.8 Å. This distance is increased to 45.0, 45.2, 45.6, and 45.2 Å for MtrR bound to PTR, TES, EST, and NDR, respectively. The RMSD for the overlay of these structures to DNA-bound MtrR are 1.10 (EST:DNA), 1.68 (TES:DNA), 0.96 (PTR:DNA), and 1.98 (NDR:DNA). The alignment of MtrR-EST and MtrR-DNA structures reveals three major movements that result in the induction of the protein (Fig S5 and Supplementary Movie 1)[46]. The first movement is a 20.8° rotation of the DNA binding domain (α1 through α3), which shifts up and away from the DNA major groove in all MtrR-steroid structures (Fig. 3B). The second significant movement is a 10° rotation and 4.5 Å shift upward of helix α4. This translation of helix α4 coincides with the upward movement of the HTH domain, away from the DNA, as well as a widening of the ligand binding pocket in the induced form. Last, the flexibility in the loop formed by residues 114–122, between helices α6 and α7, allows a 17.2° outward rotation in helix α7. This movement also aids in the enlargement of the ligand binding pocket as well as a movement of residue W136, which is positioned to obstruct the entrance to the ligand binding pocket in the DNA bound form (Fig S3). Previously the structure determination of an "apo" structure of MtrR revealed that it took the identical induced conformation likely due to the presence of CAPS, a component of the crystallization buffer, in the ligand binding pocket[18]. Superposition of this "apo" structure and the

steroid-bound structures results in an RMSD of 0.35 Å (apo-EST), 0.36 Å (apo-TES), 0.28 Å (apo-PTR), and 0.54 Å (apo-NDR).

## Ligand binding to MtrR

To confirm the modes of binding of each steroid in the ligand pocket, we made single point mutations in the binding pocket of several residues and tested the ability of each substituted protein to bind each steroid (Fig. 4, S6). Residue D171 on helix α6 appears particularly important for binding and ligand specificity because its carboxy side chain was positioned within proximity to the hydroxy groups on the five-membered ring of TES, EST and NDR. The point mutation D171A was made and binding to each steroid was determined using ITC. Additionally, based on our previous results that showed residues W136 and R176 were important for bile salt binding[18], we made and tested the binding properties of two additional mutants, W136A, and R176E. We also tested MtrR W136L, and a Q133A substitution because of the latter's observed side chain flexibility that allowed the carboxamide group to interact with both testosterone and progesterone (Fig S6A). Of all mutations, D171A affected MtrR binding most significantly resulting in a 2.9-fold increase in $K_d$ for progesterone binding, a 7.0-fold decrease in testosterone affinity, and no observable binding to β-estradiol or ethinyl estradiol (Fig. 4, S6A, B). Thus, this residue is important for ligand specificity, which is further buttressed by the finding that neither cholesterol nor cortisol, which do not have a polar atom in a similar position, bind to MtrR (Fig S1). The MtrR W136A mutant bound β-estradiol and testosterone similar to wild type but did not bind progesterone. We have posited that progesterone may take on multiple binding modes in the ligand binding pocket of MtrR, based

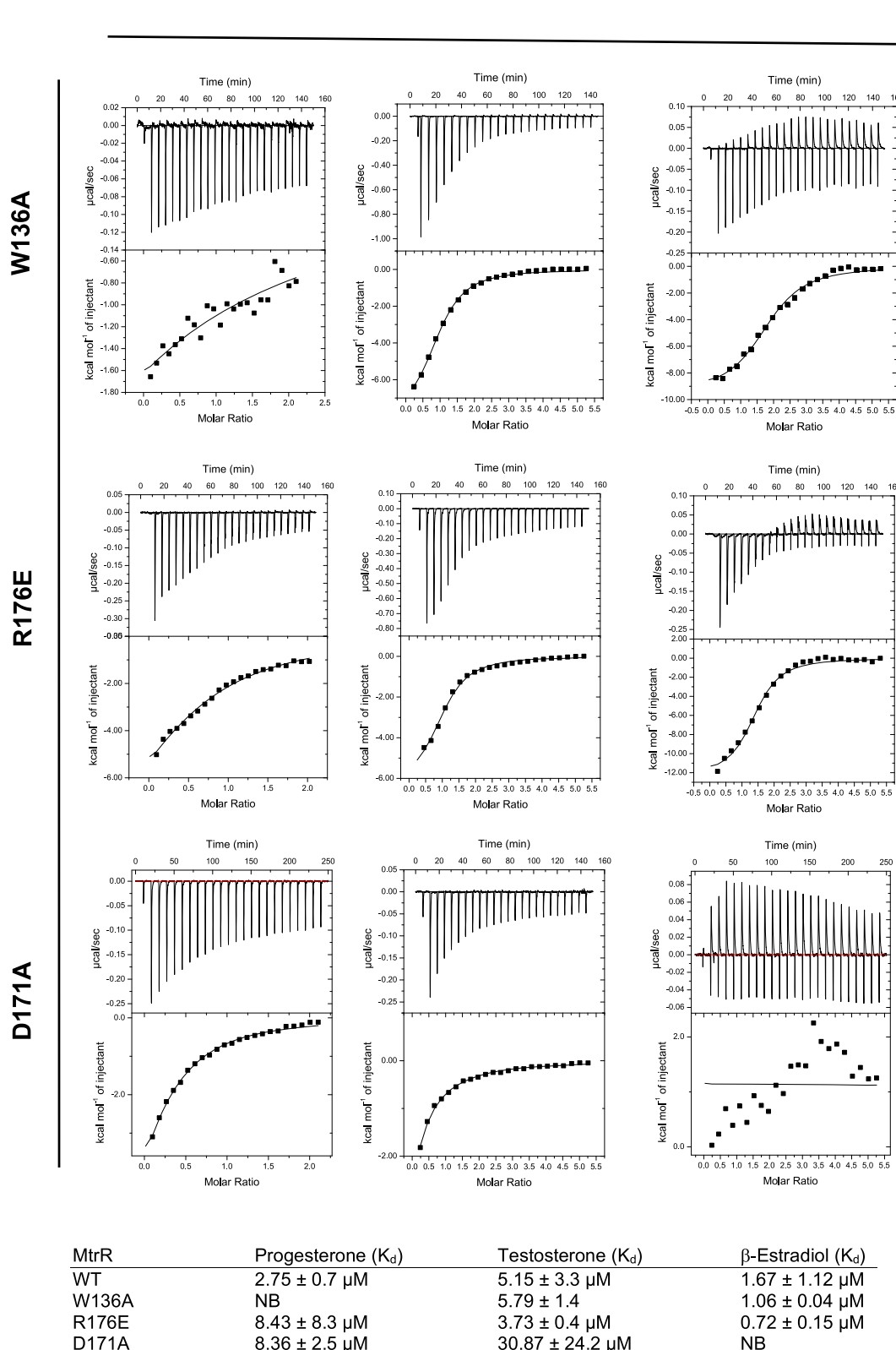

| MtrR | Progesterone ($K_d$) | Testosterone ($K_d$) | β-Estradiol ($K_d$) |
|------|---------------------|---------------------|---------------------|
| WT | 2.75 ± 0.7 µM | 5.15 ± 3.3 µM | 1.67 ± 1.12 µM |
| W136A | NB | 5.79 ± 1.4 | 1.06 ± 0.04 µM |
| R176E | 8.43 ± 8.3 µM | 3.73 ± 0.4 µM | 0.72 ± 0.15 µM |
| D171A | 8.36 ± 2.5 µM | 30.87 ± 24.2 µM | NB |

**Fig. 4 | Characterization of the MtrR ligand binding pocket.** Isothermal titration calorimetry thermograms and resulting binding isotherms for binding reactions between MtrR W136A, R176E, or D171A with progesterone, β-estradiol, and testosterone.

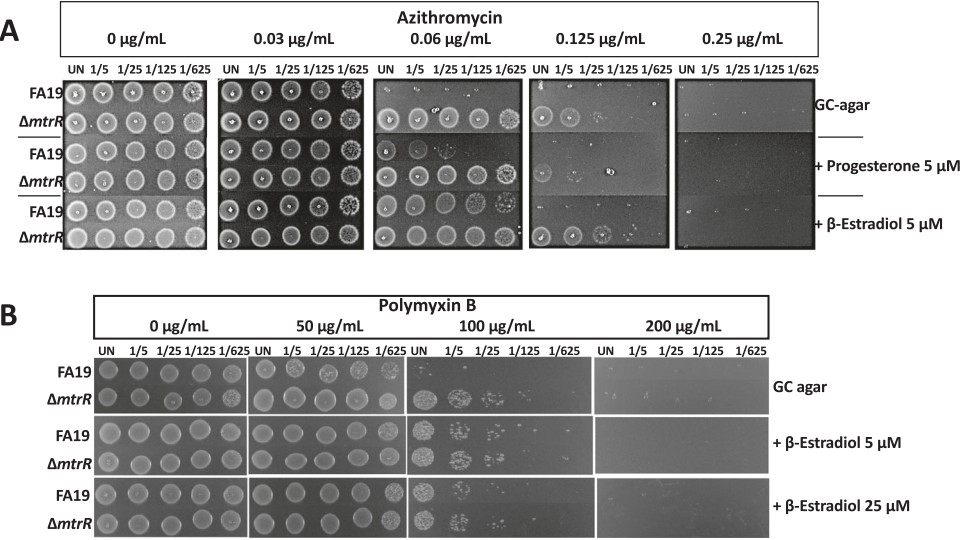

**Fig. 5 | Effect of steroidal hormones on gonococcal antimicrobial suscept-ibility.** Cultures (5 × 10⁵ CFU) and 1:5 serial dilutions of wild type FA19 and mutant strain JF1 (Δ*mtrR*) were spotted on GC agar plates containing progesterone or β-estradiol and azithromycin **A** or polymyxin B **B**. Antimicrobial susceptibility was assessed visually at 24 h.

on our crystallographic data. This may be the reason why this W136A mutant tolerates binding of testosterone, β-estradiol or ethinyl estra-diol, but not progesterone. Similarly, the $K_d$ of MtrR W136L to pro-gesterone is increased by 40-fold compared to WT but binds to β-estradiol and testosterone with comparable affinity to WT MtrR (Fig S6). MtrR R176E binds testosterone and β-estradiol with similar affinity to WT MtrR but binds progesterone with an increased $K_d$ of ~3 fold (Fig S6). MtrR Q133A bound all ligands with $K_d$ values comparable to WT MtrR. Intriguingly, MtrR Q133A bound to progesterone in an endothermic reaction, unlike all other binding events, which are exo-thermic (Fig S6).

### Hormonal effects on gonococcal cells

Based on the capacity of the above-described ligands to bind MtrR and abrogate the formation of a complex with the *mtrCDE* operator sequence we tested whether sub-lethal levels of selected hormones could induce gonococcal resistance to antimicrobials in an MtrR-dependent fashion. First, we determined the minimum inhibitory concentration (MIC) of the steroidal hormones in the gonococcal strain FA19 (Table S1) and a hormone concentration range that does not inhibit the gonococcal growth rate in liquid culture (Fig S7). We found that the MICs for PTR, EST, and TES are 127, >1175 and 555 μM respectively. Deletion of *mtrD* encoding the inner membrane compo-nent of the efflux pump lowered the MIC for PTR, EST, and TES, which indicated that these gonadal hormones are substrates of the MtrCDE pump (Table S1). This is in accord with a previous study that reported the decreased MIC for PTR in the same Δ*mtrD* strain[38]. In this regard, a concentration of up to 50 μM for each hormone did not retard the growth rate when gonococci were grown in GC broth (Fig S7). Thus, we found that in the presence of sub-lethal levels of PTR and EST wild-type strain FA19 could express elevated levels of resistance to azithromycin, a known substrate for the MtrCDE efflux pump[38], while a genetic derivative lacking MtrR did not show change in antibiotic resistance (Fig. 5A). Intriguingly, we show that MtrR does not bind azithromycin directly (Fig S1), suggesting an alternative mode of action by which MtrR induction contributes to azithromycin resistance. One possibility under investigation is that antibiotics can promote formation of reactive oxygen species that contribute to cellular death[47,48]. As men-tioned, MtrR also represses directly the expression of *rpoH*, which encodes an oxidative stress response sigma factor that is critical in the defense against reactive oxygen species (ROS) present at gonococcal

infection sites[21–23]. Thus, MtrR may be sensing ROS produced as a consequence of antibiotic treatment to induce the efflux pump. Similarly, when these strains were exposed to EST, strain FA19 dis-played increased resistance to the model cationic antimicrobial pep-tide polymyxin B (Fig. 5B).

To ascertain if the elevated antimicrobial resistance (AMR) expressed by strain FA19 in the presence of sub-lethal levels of EST and PTR was due to increased expression of *mtrCDE* we examined the in vivo expression of *mtrC* and an additional MtrR-repressed gene (*rpoH*). For this purpose, we used strains producing a wild-type MtrR, MtrR-negative strains and genetic complementation strains expressing wild-type MtrR and mutant MtrR D171A from a copy of the *mtrR* promoter and translational signals located in trans in the chromosome. We found that in the presence of increasing levels of EST, PTR, and TES expres-sion of *mtrC* and *rpoH* could be induced in the complemented (MtrR + ) strains but this induction of gene expression was abrogated due to loss of MtrR or to the production of the MtrR-D171A mutant protein (Fig. 6, S8). Importantly, genes not subjected to MtrR regulation (*lptA* and *ngo1249*) were not induced by these ligands (Fig S9). In addition to the MtrR-D171A variant, similar results were obtained with FA19 strains producing the MtrR mutant proteins with single amino acid changes at positions 136 or 176 when grown in the presence of TES, EST or PTR (Fig S10). We observed less induction by the MtrR R176E mutant compared to the MtrR W136L (Fig S10). While MtrR W136L shows almost comparable levels of activation to MtrR WT in the presence of β-estradiol (for either *mtrC* and *rpoH*) the R176E mutant shows a clearly weaker activation with β-estradiol for both genes (Fig S10). In the presence of progesterone, both mutants show almost no activation for either *mtrC* or *rpoH*, consistent with weak binding reported with ITC (Fig. 4, S6). In the presence of testosterone, MtrR W136L shows a weaker activation for *mtrC* and comparable activation to the MtrR WT for *rpoH*, while MtrR R176E shows significantly less activation for both genes (Fig S10). Lastly, we found that ethinyl estrogen could induce expression of *mtrC* and *rpoH* but only when the exposed strain pro-duced a wild-type MtrR (Fig S11).

## Discussion

*N. gonorrhoeae* is a human pathogen and the etiological agent of the sexually transmitted infection gonorrhea. Many globally isolated strains of *N. gonorrhoeae* are now multidrug-resistant, and thus has been identified as an urgent public health threat by the CDC[7].

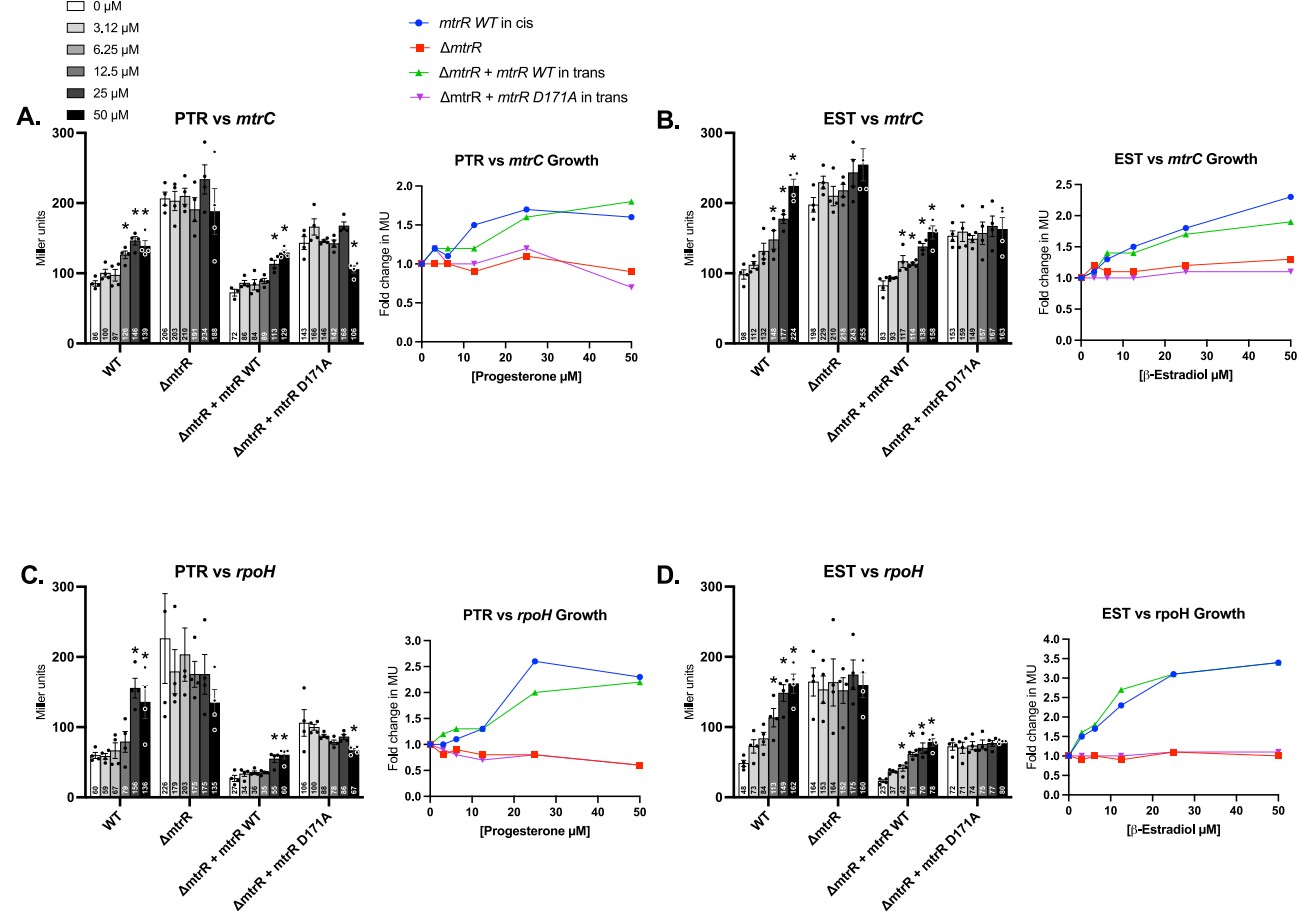

**Fig. 6 | Steroidal hormones induce the expression of gonococcal genes *mtrC* and *rpoH* in an MtrR-dependent manner.** $10^8$ CFU/mL of wild-type (WT) *mtrR* bearing strains (JC106 and JC108), *mtrR* deletion mutants (JC107 and JC109) and complemented strains in trans with WT *mtrR* (JC89 and JC91) and mutant allele *mtrR* D171A (JC101 and JC102) were grown in GC-broth to late exponential phase in increasing concentrations of progesterone (PTR, **A**, **C**) or β-estradiol (EST, **B**, **D**), statically in 96-well plates (37 °C, 5% v/v $CO_2$). Expression of *mtrC* (**A**, **B**) and *rpoH* (**C**, **D**) was measured from transcriptional and translational *lacZ* fusions and expressed as a corresponding β-galactosidase activity in Miller units (MU) as described in the Methods. Data are represented by the mean (bar) + SEM (error bar) of $n = 4$ biological samples in the bar graphs and as fold change in MU relative to the zero-hormone control in the line graphs. MU values are annotated within each bar. The legends shown for the bar and line graphs in A apply to all graphs in **C**, **D**. The experiments were performed at least thrice with reproducible results. Statistics: ANOVA test and a Dunnett's Multiple Comparison post-test (*statistically different from 0 μM at $p < 0.05$).

Understanding the mechanism(s) of MDR in pathogenic bacteria such as *N. gonorrhoeae* is paramount to combating the prospect of untreatable infections. The overexpression of multidrug efflux pumps that can export myriad antibiotics and other cytotoxic molecules from the cell is an important mechanism for *N. gonorrhoeae* survival and its development of MDR. By tightly regulating the expression of these efflux pumps, the bacteria can respond quickly to drug treatment or attacks by the host whilst minimizing the energetic cost of efflux pump biosynthesis. Here, we identify the gonadal steroid hormones testosterone, β-estradiol, and progesterone, as well as ethinyl estrogen, a component of some hormonal birth control pills, as physiologically relevant inducers of the MtrCDE efflux and RpoH stress response systems through their binding to MtrR. Although the serum concentrations of these gonadal steroids is in the nM range[49,50], the concentrations of progesterone and β-estradiol are significantly higher, low μM range, in the female reproductive tract at certain stages of menses[27]. Thus, it is likely that the local concentration of testosterone is similiarly higher at the sites of infection where production of these steroids occurs. Previous in vitro work has shown that these steroids arrest gonococcal growth[39]. During gonococcal colonization of the urogenital tract, this bacterium will encounter potentially toxic local concentrations of these steroidal hormones but which induces the MtrCDE and RpoH systems to allow its survival by efflux and upregulation of stress response genes.

This interbacterial communication between the host-produced hormones and bacterial transcriptional regulators is consistent with other systems[34]. For example, the locus of enterocyte effacement (LEE) pathogenicity island in Enterohemorrhagic *Escherichia coli* can be activated by the mammalian hormone epinephrine[29]. The presence of estradiol significantly increases the ability of *Chlamydia trachomatis* to adhere to HeLa cells[32], and progesterone increases susceptibility to infection in female rat models[33]. β-estradiol replacement in gonadectomized female and male rat models was reported to have protective effects against *Vibrio vulnificus* infection[31]. In prostate cell models, uropathogenic *E. coli* infection was surpressed by testosterone[51].

Interestingly, an early report by James and Swanson revealed phenotypic differences between gonococcal strains that were isolated from human female and male patients, as well as from strains isolated from women at different times during the menstrual cycles[52]. They showed that strains isolated from female patients tended to have higher resistance to trypsin growth inhibition[52]. Trypsin susceptibility greatly depended on the phase of the menstrual cycle when isolates were collected, with lowest resistance occuring during the luteal phase

when the concentration of progesterone is highest[52]. This study highlights the effects of different hosts environments on *Neisserial* biology[52]. Hormonal changes during the menstrual cycle influences the protective immune response in the female reproductive tract to optimize conditions for defending against pathogen infection and also repressing immune reponses whilst remodeling tissue to enable implantation and pregnancy[26,27]. This creates a "window of vulnerability" wherein pathogens can enter and infect the reproductive tract when immune protection is dampened in order to optimize pregnancy[26,27]. This is consistent with our findings that these gonadal steroids induce stress responses in gonococcal cells that are invading and colonizing a highly dynamic environment.

The structures of MtrR bound to these steroids reveal the ligand-recognition and induction mechanisms and structural consequences of ligand binding. These structures elucidate the highly hydrophobic/aromatic binding pocket of MtrR and the presence of a key aspartate residue that accommodates steroid binding. Site directed mutagenesis of selected residues within the ligand binding pocket reveal varying effects on binding. MtrR D171 is perhaps the most critical contributor to hormonal steroid binding. Additionally, the interaction between D171 and a polar atom of each steroid appears to be a key determinant in discriminating against other steroids such as cholesterol and cortisol, which lack an appropriate proximal polar group. Such a binding pocket is consistent with other multiligand binding proteins such as QacR from *S. aureus*[35], TtgR from *Pseudomonas putida*[53], CmeR from *Campylobacter jejuni*[54] and AcrR from *E. coli*[55], the drug-binding sites of which consist mostly of aliphatic and aromatic residues with only a few, but essential, polar residues.

The structures of the MtrR-steroids complexes represent the induced form of MtrR as the center-to-center distance between the recognition helices of the helix-turn-helix DNA binding motifs is not suitable for specific recognition of the nucleobases of two consecutive major grooves when compared to the DNA bound conformation of MtrR[25]. Such an increase in this distance is consistent with other induction mechanisms of TetR family members[35,56–59]. Although members of the TetR family show strong structural homology in their DNA binding domains, the ligand binding pockets vary in composition and volume. For example, the ligand binding pocket of QacR, is unusual for a multiligand binding protein as it utilizes glutamates that become available upon ligand binding due to a conformational change, as well as polar residues to facilitate the binding of multiple drugs, but maintains critical aromatic interactions with each compound[35]. Binding of bile salts to RamR is facilitated by four hydrogen bonding residues, whilst relying only on π-π stacking to bind compounds such as dequalinium, ethidium, and rhodamine 6G[44,60]. Here, we show that MtrR utilizes only one polar residue, D171, to bind specifically to gonadal steroids. Like QacR, MtrR recruits a key surface-exposed residue, W136, upon ligand binding thereby allowing the indole ring to cap the pocket[35].

In conclusion, we have elucidated the binding and induction mechanisms of hormonal steroids by MtrR, a transcriptional regulator that is a key player in the multidrug resistance arsenal of *Neisseria gonorrhoeae*. Significantly, the induction of the *mtrCDE* efflux genes and the gene encoding the *rpoH* stress response sigma factor by ethinyl estrogen suggests that steroid-based hormonal contraceptives might be co-opted by *Neisseria gonorrhoeae* to help colonize its human host, hence underscoring that, although these medicines are highly effective in preventing unwanted pregnancies, they do not protect against gonococcal infection.

## Methods

### Protein overexpression and purification

The gene encoding *mtrR* is codon optimized (GenScript) and synthesized for expression in *E. coli*. The gene was subsequently subcloned into the ampicillin resistant pMCSG7 vector in-frame with a T7 pro-motor, and contains an N-terminal Hexa-histidine tag and a tobacco etch virus (TEV) protease cleavage site. Rosetta Gami B(DE3) pLysS cells were transformed by the plasmid for protein expression. The cells were grown in Luria-Bertani broth containing ampicillin, chloramphenicol, and tetracycline at 37 °C. Cultures were grown to an optical density ($OD_{600}$) of 0.5–0.6 and expression of MtrR was induced by the addition of 1 mM isopropyl-β-D-thiogalactopyranoside (IPTG) and grown for an additional 3 h before harvesting. Cells were centrifuged, reconstituted in buffer containing 20 mM Tris-HCl pH 8.0, 200 mM NaCl, 10% glycerol, and 1 mM tris-2-carboxyethyl phosphine hydrochloride (TCEP) and lysed by sonication. MtrR was purified from clarified lysate by $Ni^{2+}$-nitrilotriacetic acid (Ni-NTA) affinity chromatography followed by cleavage of the Hexa-histidine tag by TEV digestion. MtrR was further purified by size exclusion chromatography (S200 column). Selenomethionine modification of MtrR was achieved using the methionine-inhibitory pathway and purified as above[61].

### Crystallization and data collection

Crystals were grown at 25 °C using the hanging drop-vapor diffusion method. MtrR was co-crystallized with 2.5 mM progesterone, β-estradiol, testosterone, or ethinyl estradiol with 5% MeOH. Crystals formed within two weeks in crystallization solutions containing 1200 mM sodium phosphate monobasic, 800 mM potassium phosphate dibasic, 100 mM CAPS/ NaOH pH 10.5, and 200 mM lithium sulfate. Crystals were cryoprotected in this buffer with 25% glycerol added for flash freezing in liquid nitrogen. Density data were collected at the Advance Light Source (ALS) on beamline 5.0.2 (MtrR-EST and MtrR-TES), 5.0.1 (for MtrR-PTR), or 5.0.3 (for MtrR-NDR). The wavelengths for data collection were 0.9763 (MtrR-EST), 0.9786 (MtrR-TES), 0.9774 (MtrR-PTR), and 0.9765 (MtrR-NDR). Data were collected at 100 K. Crystals of selenomethionine-substituted MtrR in complex with β-estradiol were used for SAD data collection. Phenix Autosol (using Phenix version 1.19)[62,63] was then employed to locate and refine Se heavy atom sites, determine initial phases, and carry out density modification. This produced an excellent experimental map, in which 76% of the structure was autotraced. Strong density for β-estradiol was observed in this map. The initial model was subjected to refinement in Phenix-refine and multiple cycles of rebuilding and refinement[62]. After the final refinement, a dimer of this structure was used in molecular replacement to solve the structures of MtrR bound to progesterone, testosterone, and ethinyl estrogen. Iterative cycles of the model building were done in COOT (version 0.9.6)[64] and refinement using Phenix refine[62,65,66]. The stereochemistry was excellent with the favored percent Ramachandran for each structure: 98.82 (MtrR-EST), 98.02 (MtrR-TES), 98.25 (MtrR-PTR), and 87.86 (MtrR-NDR.) There were no Ramachandran outliers in any structure except for MtrR bound to NDR which had the lowest resolution. Clash scores for all structures were less than or equal to 6 except MtrR bound to NDR, which had a clashscore of 10.83. Mean I/sigma(I) values for the highest resolution shell were 0.7 (MtrR-EST), 2.2 (MtrR-TES), and 2.3 (for both MtrR-PTR and MtrR-NDR). Selected data collection and refinement statistics are listed in Table 1. Alignment analysis and protein visualization were done in Pymol (version 2.5.4).

### Isothermal titration calorimetry

Purified MtrR was concentrated to 20 µM in purification buffer plus 1% MeOH. The addition of 1% methanol is necessary to dissolve each hormonal steroid, cortisol, and cholesterol in the purification buffer. Titrations with 20 or 5 µM MtrR in the sample cell and 125, 200, or 500 µM ligand in the syringe were performed using a VP-ITC microcalorimeter (Microcal Inc.). ITC experiments were conducted at 25 °C with a stirring speed of 199 rpm. Spacing between injections was 600

**Table 1 | Data collection and refinement statistics**

|  | MtrR + β-Estradiol | MtrR + Testosterone | MtrR + Progesterone | MtrR + Ethinyl Estradiol |
|---|---|---|---|---|
| **Data collection** |  |  |  |  |
| Space group | C 1 2 1 | I 1 2 1 | I 1 2 1 | I 1 2 1 |
| Cell dimensions |  |  |  |  |
| $a, b, c$ (Å) | 219.31, 84.60, 59.11 | 58.91, 84.87, 212.52 | 58.18, 85.05, 211.81 | 57.92, 85.71, 211.56 |
| a, b, γ (°) | 90, 103.1, 90 | 90, 92.25, 90 | 90, 91.24, 90 | 90, 90.78, 90 |
| Resolution (Å) | 46.46–2.37 (2.46–2.37) | 48.37–2.22 (2.29–2.22) | 48.01–2.40 (2.40–2.31) | 43.52–3.2 (3.314–3.2) |
| $R_{merge}$ | 0.064 (0.226) | 0.049 (0.747) | 0.031 (0.377) | 0.051 (0.368) |
| $I$ / sigma($I$) | 10.7 (0.7) | 17.3 (2.2) | 14.6 (2.3) | 11.53 (2.3) |
| Completeness (%) | 95.52 (99.7) | 99.9 (99.9) | 98.1 (100) | 98.46 (97.3) |
| Redundancy | 5.3 | 7.3 | 3.3 | 1.9 |
| CC1/2 | 0.999 (0.979) | 0.999 (0.850) | 0.999 (0.886) | 0.997 (0.815) |
| **Refinement** |  |  |  |  |
| Resolution (Å) | 2.37 | 2.22 | 2.31 | 3.20 |
| No. reflections | 40975 (3907) | 51794 (4520) | 44455 (4492) | 16981 (1655) |
| $R_{work}$ / $R_{free}$ (%) | 19.2/24.4 | 20.8/26.1 | 21.0/25.1 | 20.1/27.7 |
| No. atoms | 6523 | 6155 | 6114 | 6125 |
| Protein | 6169 | 5979 | 5955 | 5993 |
| Ligand/ion | 116 | 109 | 108 | 118 |
| Water | 238 | 67 | 51 | 14 |
| $B$-factors | 41.38 | 62.11 | 66.12 | 87.67 |
| Protein | 40.94 | 62.05 | 65.58 | 87.41 |
| Ligand/ion | 61.86 | 67.17 | 99.26 | 102.13 |
| Water | 42.87 | 59.21 | 59.03 | 76.62 |
| r.m.s. deviations |  |  |  |  |
| Bond lengths (Å) | 0.008 | 0.008 | 0.007 | 0.009 |
| Bond angles (°) | 1.02 | 0.93 | 0.82 | 1.15 |
| Ramachandran Analysis |  |  |  |  |
| Favored (%) | 98.8 | 98.0 | 98.3 | 87.9 |
| Allowed (%) | 1.2 | 2.0 | 1.7 | 10.8 |

*Values in parentheses are for highest-resolution shell.

or 300 s (no difference in results found). Germane blanks were subtracted, and the data analyzed using ORIGIN 7.0[67]. At least three experimental measurements, technical and some biological replicates, were averaged for each reported binding constant.

**Fluorescence polarization based-DNA binding assays**

Fluorescence polarization based-DNA binding data were collected using a Panvera Beacon 2000 fluorescence polarization system (Invitrogen) and analyzed with Prism (Graphpad Software 9, 10). Purified MtrR was buffer exchanged into "DNA" binding buffer (150 mM NaCl, 20 mM Tris-HCl pH 7.5, 2.5% glycerol, 1 mM TCEP). MtrR was titrated into a binding buffer solution that contained 1 nM 5'-fluoresceine-labeled DNA (containing either the *rpoH* or *mtrCDE* operator site), 1 μg of poly(dI-dC) DNA, 1 μg bovine serum albumin (BSA) and 125 μM progesterone, testosterone, or β-estradiol (or ethinyl estradiol, see Fig. S2) resulting in a final methanol concentration of 1%. The reported oligoduplexes containing the *rpoH* and *mtrCDE* binding site differ in lengths, 27mer and 21mer, respectively[17,21]. Previous work from our laboratories has shown that these differences in length have no impact on binding affinity between the two sites[25]. BSA and dI-dC are included as carriers and to control for nonspecific DNA binding. Samples were excited at 490 nm and polarized emissions measured at 530 nm. Controls were run with no protein (DNA ± methanol, hormonal steroids) to ensure that millipolarization values were not affected by experimental conditions alone. The binding affinity of protein for DNA was determined using the following equation: $P = P_f + (P_b - P_f) [M] /$ $(K_d + [M])$ where P is the measured polarization, $P_f$ is the polarization of free DNA, $P_b$ is the polarization of the completely bound DNA, $K_d$ is the equilibrium dissociation constant, and M is the concentration of protein. $P_b$ and $K_d$ were determined by nonlinear least-squares analysis. At least three independent experimental measurements were averaged for each reported binding constant.

**Strains and media**

*N. gonorrhoeae* strains used in this study are described in Table S2. Gonococcal cultures were grown overnight at 37 °C in a 5% (v/v) $CO_2$ incubator on GC agar (BD Difco) plates supplemented with Kellogg's supplements I and II[68]. Growth in liquid medium was at 37 °C in GC broth (Proteose Peptone No.3 15 g/L, dipotassium phosphate 4 g/L, monopotassium phosphate 1 g/L, sodium chloride 5 g/L) containing Kellogg's supplements I and II and 0.042% (w/v) sodium bicarbonate. *E. coli* TOP10 cells (Life Technologies, Carlsbad, CA) were used for cloning steps and grown on LB medium[69].

**Gonococcal genetic transformations**

Plasmids and oligonucleotide primers used for gonococcal genetic transformations are described in Table S2 and S3 respectively. Genetic transformation of *N. gonorrhoeae* was achieved by homologous recombination using the spot agar transformation method described perviously[70]. A complete description of cloning and transformation steps of derivatives of gonococcal strain FA19 is described in the supplementary material.

## Growth of gonococcal strains in 96-well plates and β-galactosidase assays

To determine the effect of the steroidal hormones on the expression of MtrR-regulated genes, gonococcal strains bearing the transcriptional and translational *lacZ* fusion constructs were grown in GC-broth in 96-well plates (Costar #3370 flat bottom, sterile, with lid). β-galactosidase activities were determined by hydrolysis of the colorimetric substrate ortho-Nitrophenyl-β-galactoside (ONPG) using a modified version of the J.H Miller[71] protocol as described in the supplementary material.

## MIC and antimicrobial susceptibility testing

To determine the minimal inhibitory concentrations (MICs) by the steroidal hormones and their effect on antimicrobial susceptibility, 5 x 10^5 CFU and 1:5 serial dilutions of gonococcal strains were spotted on GC agar plates containing different concentrations of antibiotics and/or hormones as indicated. Antimicrobial susceptibility was assessed after 24 h incubation.

## Site directed mutagenesis

The gene encoding the MtrR protein was codon optimized and synthesized for expression in *E. coli* (GenScript). The gene was subsequently subcloned into the ampicillin resistant pMCSG7 vector in-frame with a T7 promotor, N-terminal Hexa-histidine tag, and tobacco etch virus (TEV) protease cleavage site. This was used as the template for site directed mutagenesis. Site directed mutagenesis was performed with complementary oligonucleotides containing the desired mutations, obtained from Integrated DNA Technologies (IDT), as previously describIbed[18]. Protein was expressed and purified as described in Materials and Methods.

## Plate-based fluorescence polarization DNA-binding assay

To analyze the effect of ethinyl estradiol on DNA binding by MtrR, fluorescence polarization DNA-binding data were collected in 96-well plates (Costar #3915 black, flat bottom). with a SpectraMax M5e Microplate Reader (Molecular Devices) and analyzed with Prism (Graphpad Software, version 10.0.0). Purified MtrR was buffer exchanged into "DNA" binding buffer (150 mM NaCl, 20 mM Tris-HCl pH 7.5, 2.5% glycerol, 1 mM TCEP). MtrR was titrated into a binding buffer solution that contained 10 nM 5′-fluoresceine-labeled DNA (encompassing either the *rpoH* or *mtrCDE* operator site), 1 μg of poly(dI-dC) DNA, 12 μg bovine serum albumin (BSA) and 125 μM ethinyl estradiol resulting in the final methanol concentration of 1%. The binding affinity of protein for DNA was determined using the Morrison's quadratic equation[72]. BSA and dI-dC are included as carriers and to control for nonspecific DNA binding. Samples were excited at 485 nm and polarized emissions measured at 525 nm, with a 515 nm cutoff. Controls were run with no protein (DNA ± methanol, ethinyl estradiol) to ensure that milli-polarization values were not affected by experimental conditions alone. At least three independent experimental measurements were averaged for each reported binding constant.

## Gonococcal genetic transformations

Plasmids and oligonucleotide primers used are described in Tables S2 and S3, respectively. To study the hormone effect on the expression of MtrR-regulated genes *mtrC* and *rpoH*, *lacZ* transcriptional and translational fusions were constructed using pLES94[73] and integrated into the *proAB* locus of *N. gonorrhoeae* strains. Genes *lptA* and *ngo1249*, not present within the MtrR regulon, were used as negative controls. To obtain the *mtrC-*, *rpoH-* and *ngo1249-lacZ* fusions the promoter region and the initial 3 to 6 codons of the genes were amplified from FA19 gDNA using the primer pairs mtrC-lacZ-F/mtrC-lacZ-R, rpoH-lacZ-F/rpoH-lacZ-R and 1249-lacZ-F/1249-lacZ-R respectively. The *lptA-lacZ* fusion was constructed before[74]. The resulting PCR products were digested with BamHI, cloned into similarly digested pLES94 and transformed into TOP10 cells. Blue colonies were selected in LB agar

containing chloramphenicol (10 μg/mL) and X-gal (20 μg/mL). The resulting recombinant plasmids were sequenced with primers proABFw and lacZRv. Plasmids bearing the expected sequences were linearized with HindIII and transformed into *N. gonorrhoeae* strains (Table S2) using a spot agar transformation method[70] and selection on GC-agar containing chloramphenicol (0.5 μg/mL). Integration of the *lacZ* fusion at the *proAB* locus was confirmed by PCR with primers proABFw and lacZRv and by Sanger sequencing.

To complement *mtrR* expression in *mtrR* null mutant strain JF1, the promoter region and coding sequence (CDS) of *mtrR* was amplified from FA19 gDNA using primers pacImtrR-Fw and pmeImtrR-Rv. The PCR fragment was double digested with PacI and PmeI and ligated into similarly digested pGCC3[75]. The resulting pGCC3-*mtrR* was sequenced with pGCC3Rv and mtrR-qRT-R and used to transform strains JC107 and JC109 to obtain strains JC89 and JC91, respectively. Selection of the transformant strains was carried out in GC-agar containing erythromycin (1.0 μg/mL). Correct integration of the plasmids into the *lctP-aspC* loci was confirmed by PCR with primers lctPqfFw and pGCC3Rv and sequencing of the PCR products with pGCC3Rv and mtrR-qRT-R.

To complement *mtrR* expression with mutant alleles encoding the missense mutations in MtrR W136L, R176E, or D171A overlap extension PCRs were prepared to fuse a WT *mtrR* promoter region with mutant *mtrR* alleles. Briefly, a PCR fragment encoding the promoter region was obtained with primers pacImtrR-Fw and mtrR-midRv and FA19 gDNA; while the mutant alleles were amplified from site-directed mutagenesis created-vectors pGAB020RGB (*mtrR* W136L), pGAB018RGB (*mtrR* R176E) and pmtrRD171A (*mtrR* D171A) using primers mtrR-mid-Fw and PmeI-mtrR-Rv. The two resulting PCR fragments (promoter + mutant *mtrR*) were mixed and used as templates for the amplification of the fused promoter region-mutant *mtrR* alleles fragments with primers pacImtrR-Fw and PmeI-mtrR-Rv. These final PCR fragments were digested with PacI and PmeI and ligated into similarly digested pGCC3. Complementation vectors expressing the mutant *mtrR* alleles from their own transcriptional and translational signals were transformed into strains JC107 and JC109 as described above for pGCC3-*mtrR*.

## Growth of gonococcal strains in 96-well plates and β-galactosidase assays

To determine the effect of the steroidal hormones on the expression of MtrR-regulated genes, gonococcal strains bearing the *lacZ* fusion constructs were grown in GC-broth in 96-well plates (Costar #3370 flat bottom, sterile, with lid). Cultures (100 μL) were adjusted to an initial optical density at 600 nm (OD600) of 0.1 containing serial dilutions of the hormones as indicated and were incubated statically at 37 °C in the 5% (v/v) CO2 incubator for 7 h until late-exponential growth phase. Stock solutions of the hormones were prepared to 3.3 mM in a 2:1 mix of methanol:$H_2O$. The OD600 of the plates was determined in a plate reader and the plates were kept in ice until use. Next, cell lysis was achieved by resuspending 18.75 μL of the cultures in 56.25 μL of CelLytic B Cell lysis reagent (Sigma) in a new 96-well plate and incubating 30 min at 26 °C. Beta-galactosidase reactions started by adding 100 μL of a reaction mix containing 62.5 μL of buffer Z (60 mM $Na_2HPO_4$, 40 mM $NaH_2PO_4$ pH 7.0, 10 mM KCl, 1 mM $MgSO_4$) and 37.5 μL of 4 mg/mL ortho-Nitrophenyl-β-galactoside (ONPG) (diluted in 0.1 M phosphate buffer pH 7.0). Reactions were incubated at 26 °C until a pale-yellow color developed, were stopped with 73 μL of 1 M $Na_2CO_3$ and the OD420 was read with a plate reader. A modified formula was used to calculate the Miller units[71] of the cultures: Miller units = $((OD_{420\,nm} - OD_{420\,nm\,blank}) \times 1.4925 \times 1000)/(t(min) \times V(mL) \times 6.857 \times OD_{600nm} \times 5.5897)$, where 1.4925 is the conversion factor to correct the OD420 obtained with the light path of a 248 μL volume in 96-well plates to the light path using cuvettes (1 cm) in a spectrophotometer, t is the incubation time of the assay in min, V is the volume of cells used for the assay (18.75 μL), 6.857 is to convert the volume of cells used in the microplate assay to the

equivalent volume used in the original Miller protocol, $OD_{600nm}$ is the optical density of the cells after growth, and 5.5897 is the conversion factor to transform the OD600 of the cultures in 96-well plates (100 μL) to the $OD_{600nm}$ obtained using a spectrophotometer and a cuvette with a light path of 1 cm. In this formula the turbidity correction ($1.75 \times OD_{550nm}$) used in the original Miller formula was omitted due to the low turbidity of the reactions.

## Reporting summary

Further information on research design is available in the Nature Portfolio Reporting Summary linked to this article.

## Data availability

All data in this study are included in the article. The coordinates and structure factors for the MtrR-PTR, MtrR-EST, MtrR-TES, and MtrR-NDR complexes have been deposited in the RCSB Protein Data Bank with the accession codes 8FW8, 8FW0, 8FW3, and 8SSH, respectively. Source data are provided in this paper.

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

## Acknowledgements

We would like to thank the beamline scientists at ALS BL 8.3.1 and BL 5.0.2 for their help with data collection. In addition, we acknowledge the Advanced Light Source at Lawrence Berkeley National Laboratory. National Institutes of Health [R35 GM130290 to M.A.S., R05 AI048593 to R.G.B., R01 AI021150 to W.M.S.]; W.M.S. is the recipient of a Senior Research Career Scientist Award from the Biomedical Laboratory Research and Development Service of the U.S. Department of Veterans Affairs; The Advanced Light Source is supported by the Director, Office of Science, Office of Basic Energy Sciences, Material Sciences Division, of the U.S. Department of Energy [DE-AC03-76SF00098]. Funding for open access charge: Duke University School of Medicine.

## Author contributions

G.M.H., J.C.A., W.M.S., and R.G.B. designed the experiments and analyzed the biochemical and cellular data. G.M.H., R.G.B., and M.A.S. analyzed the structural data. G.M.H. and G.A.B. generated purification constructs. G.M.H. purified proteins, determined their structures, and performed biochemical characterizations. M.A.S. and G.M.H. collected X-ray crystallography data and M.A.S. provided X-ray crystallography consulting and experimental input. G.M.H. J.C.A., W.M.S., and R.G.B. wrote the manuscript. J.C.A., C.L.H., and V.D. conducted cellular experiments. We also thank J.R.P. for insights into steroid and pathogen biology and for providing testosterone for the project. All authors have read and approved the manuscript.

## Competing interests

The authors declare no conflict of interest.
