## [Peer Review File · Nature Communications]

Hormonal Steroids induce multidrug resistance and stress response genes in *Neisseria gonorrhoeae* by binding to MtrRReviewer #1 (Remarks to the Author):

The manuscript by Hooks et al. describes the studies on the MtrR regulator related to stress response in *Neisseria gonorrhoeae*. The manuscript is well-written. The structural and biochemical data are well-presented. The authors solved four crystal structures of the MtrR-ligand complexes and measured the dissociation constants for the complexes. They used many experimental techniques and proved that binding of the analyzed steroids results in decreased affinity of MtrR for cognate DNA. Hence, these steroids could be physiological ligands for the regulator which is an important finding. The TetR family consists of more than 0.5 million members and the ligand-binding domains are very variable. The structural and biochemical characterization of the particular regulator significantly contributes to the proper identification of its function. Therefore, I recommend publishing this article after a few minor corrections listed below.

- 1) The authors used the rpoH 27mer and mtrCDE 21mer, it should be indicated why such lengths of DNA fragments were used.
- 2) Lines 222-232: The authors described three major structural differences between the MtrR-ligand and MtrR-DNA complexes, the Fig. 3 should include three additional panels, each showing the indicated differences.
- 3) Line 229: "the loop formed by residues 114-122 allows" – between which helices?
- 4) Table 1: the statistics of the Ramachandran plot should be included.

Reviewer #2 (Remarks to the Author):

The manuscript by Hooks et al. reports the structures of MtrR, the transcriptional regulator of the MtrCDE efflux pump as well as several other genes/proteins in *N. gonorrhoeae*, bound to several steroid hormones, and interrogates the structural implications of hormone binding. The data show that MtrR directly binds progesterone, estradiol, and testosterone (and ethinyl estradiol found in some birth control pills) and "activates" MtrR (i.e. causes release from its DNA binding sites), and that this activation can increase antibiotic resistance to other antibiotics or host antimicrobial peptides.

Overall, I found the evidence to be very convincing, the various approaches used to be appropriate and complementary, and believe the study illuminates fundamentally important aspects of gonococcal host-pathogen interactions. That said, there were areas where the presentation could be improved. Below are many comments (most are minor) from a careful reading that would help clarify the presentation.

1. The stoichiometries of binding for progesterone and estradiol in ITC experiments (and comparison to structural data) are confusing. For example, the stoichiometry of progesterone is reported as 0.5, but at least one of the 2 dimers in the asymmetric unit of the crystal structure shows each monomer with a bound progesterone (the other one has CAPS bound in one monomer, presumably from the crystallization solution) (Fig 2 and lines 176-177). Why is it different than the ITC experiments? The explanation provided does not address these contradictory data satisfactorily.
2. In Fig. 2, the estradiol binding experiments seems to show a molar ratio of <0.5, despite the $N \sim 1$ next to the graph. In fact, the highest molar ratio shown on the X axis is 0.5
3. In Fig. 4, the 3 mutants appear to have a stoichiometry of binding for progesterone closer to 1 than 0.5.
4. It took me a while to understand the structural changes that occur in MtrR upon binding of steroid hormones (Fig 2B), in part because it looks like the HTH in the subunit on the left is pretty well aligned with that of the apo structure whereas the right side subunit is not. I was expecting to see the HTH motifs of the two subunits to be equally displaced from the apo structure. Is this a feature of the overlay procedure? Are the conformational changes in the two subunits identical? If they are, perhaps a more balanced overlay (in which the conformational changes are shown equally for the two subunits) would help make this clearer.
5. Line 136, the FP experiments of testosterone- and estradiol-bound MtrR show a 3-fold decrease in binding affinity for the mtrCDE operator (progesterone causes even less of a decrease, 2.3).

These fold-decreases in affinity seem very small. Presumably it is sufficient to allow induction of transcription, but perhaps the authors could comment and explain why such a small shift in affinity is physiologically relevant. Could this binding be influenced by the presence of RNA polymerase in the full promoter DNA?

6. Lines 201-203 discuss residues (K167, R176', and R137) that coordinate a phosphate group in MtrR. It seems to me that if they are interesting enough to mention, that they be shown in Fig 3C. Also, does phosphate (or similar ion) have a role in MtrR function, e.g. in stabilizing the MtrR dimer, or is it simply there from crystallization buffer, which is high in phosphate?

7. Lines 251-252 has poor sentence construction: "which is further buttressed by the finding that neither cholesterol nor cortisol, which do not have a polar atom in a similar position, do not bind to MtrR (Fig S1). Neither x nor y...do not bind

8. Lines 270-272: "Deletion of mtrD encoding the inner membrane component of the efflux pump significantly lowered the MIC for PTR, EST, and TES, which indicated that these gonadal hormones are substrates of the MtrCDE pump (Table S1)." I would suggest removing "significantly", since no statistics are shown and except for Testosterone, the fold change is either 2-fold (considered not significant) or unknown (estradiol).

9. The authors report no binding (NB) of progesterone to the W136A mutant (Fig. 4) but do report a binding affinity to the W136L mutant (Fig S5). However, when I look at those ITC graphs, the W136A mutant data (e.g. point to point variation) looks similar if even better in W136A than W136L, and also has a curve fit-why no Kd?

10. The authors should include WT binding data in the table in Fig S5.

11. While I agree that adding a sub-MIC level of steroid hormones decreases the capacity of Azi and Polymixin B to inhibit GC growth, the magnitude of this effect is unclear to me. Do the MICs of the two antibiotics change? How do the dilution experiments translate to resistance?

12. Lines 296-297, "Importantly, genes not subjected to MtrR regulation (lptA and ngo1249) were not induced by these ligands (Fig S8)." Lines 462-463. "Steroidal hormones do not affect the expression of genes outside the MtrR regulon." Using 2 genes to imply that these hormones don't affect expression outside of the MtrR regulon seems an overreach. This really can't be answered without an RNAseq experiment. I'm not suggesting that one be done, just that the text be modified to reflect this limitation.

13. The data visualization in Fig 6 (as well as S7, S9, and S10), particularly the line graphs, nicely shows the effects of the mutations. These data are convincing that the D171A mutant does not respond to steroid hormones.

14. How specific is the D171A mutation for preventing hormone-promoted activation in MtrR? Do other non-steroidal inducers still activate MtrR?

15. Lines 298-300, "In addition to the MtrR-D171A variant, similar results were obtained with FA19 strains producing the MtrR mutant proteins with single amino acid changes at positions 136 or 176 when grown in the presence of TES, EST or PTR (Fig S9)." This statement does not comport with the data shown. While I agree that the D171A mutant is not functional, the data with the other mutants are mixed. By my eye, I ascertained the following from the data in Fig 4, S7, S9, and S10:

D171A. No activation=>Estradiol, Progesterone, Testosterone, Ethinyl estradiol

W136A. Not done

W136L. Activation=> Estradiol, Testosterone (rpoH); Weak activation=> Testosterone (mtrC)

R176E. Activation=> Estradiol (rpoH); Weak activation=> Testosterone (rpoH), Progesterone (rpoH), Estradiol (mtrC); No activation=> Progesterone (mtrC)

The authors in fact don't really discuss the mutants sufficiently, although they show the data in the supplemental figures. This should be addressed.

Reviewer #3 (Remarks to the Author):

This manuscript describes analysis of the function of the MtrR regulatory protein. Using a number of in vitro assays, together with determining high resolution structures, insight has been provided

into the identity of steroid compounds that can act as inducers of the system and which instigate a conformational switch in the regulator. The Mtr system is not just involved in the regulation of the mtrCDE operon but can act as a master regulator affecting multiple different factors both in a positive and negative fashion.

The authors have previously published work on MtrR, including structures bound to two regulatory sequences and a pseudo apo structure. In the latter, a detergent molecule was found in the binding site and therefore the protein was in the induced configuration.

Identification of compounds that can induce the expression of a multidrug export protein can aid in discerning the function and biological importance of the export system, thus these findings are of interest.

It is intriguing to see different stoichiometries of drugs bound to MtrR. Three ligand bind with one ligand:protomer, while the ligand PTR appears to be 1:dimer. It is worded strongly that 1 PTR is unambiguously bound to one dimer however in the next sentences there is a caveat describing the possibilities that a CAPS molecule could be present or a second PTR. Another alternative that was posed is that a progesterone molecule may be in the asymmetric unit without explaining how this could occur. The wording could therefore be changed as these data are convincing as with the other ligands and the explanation for the interactions are not as clear cut.

Azithromycin, itself a substrate of the MtrCDE pump, doesn't not appear to bind to MtrR and therefore does not induce expression of the efflux system. An alternative hypothesis of the way the presence of the drug may still be recognised by the pump has been presented. It is probably beyond the scope of this manuscript but it will be interesting in future to decipher this different mechanism.

Specific corrections.

Line 18 and 69 - Multiple transferable resistance Repressor should read multiple transferable resistance repressor

Line 34 - gram-negative should read Gram-negative

Line 112 - should read a significant decrease.

Line 160 -162 is a repetition of previous findings and could be deleted/made more succinct.

Line 245 - W137A should be W136A

Line 246 - W137L should be W136L

Line 295 - Using the nomenclature MtrR+ could signify overexpression of MtrR protein. If this is not what was intended, it would be better to change this to WT or to signify if it was a complemented strain.

Line 304 - Neisseria should be N.

Line 327 - th should be the

Line 355 - Staphylococcus should be S.

Line 356 - Escherichia should be E.

Line 513 - should it be and azithromycin?

Response to Reviewer Comments

We would like to thank the reviewers for their positive reviews, helpful critiques, and suggestions. We are excited about this manuscript and appreciate the time and thoughtfulness of your consideration. Following are our point-by-point responses to each review.

Reviewer #1 (Remarks to the Author):

The manuscript by Hooks et al. describes the studies on the MtrR regulator related to stress response in *Neisseria gonorrhoeae*. The manuscript is well-written. The structural and biochemical data are well-presented. The authors solved four crystal structures of the MtrR-ligand complexes and measured the dissociation constants for the complexes. They used many experimental techniques and proved that binding of the analyzed steroids results in decreased affinity of MtrR for cognate DNA. Hence, these steroids could be physiological ligands for the regulator which is an important finding. The TetR family consists of more than 0.5 million members and the ligand-binding domains are very variable. The structural and biochemical characterization of the particular regulator significantly contributes to the proper identification of its function. Therefore, I recommend publishing this article after a few minor corrections listed below.

1) The authors used the *rpoH* 27mer and *mtrCDE* 21mer, it should be indicated why such lengths of DNA fragments were used.

The oligoduplex sequences that we used are from our previous data identifying the MtrR binding sites in the *rpoH* and *mtrCDE* promoters. The lengths we used are based on our biochemical experiments which identified the minimal sequences needed for high affinity binding by MtrR to these operator sites (see Beggs et al., 2021). An explanation has been added to the FP methods section: "The reported oligoduplexes containing the *rpoH* and *mtrCDE* binding site differ in lengths, 27mer and 21mer, respectively^{1,2}. (Lines 445-448)

2) Lines 222-232: The authors described three major structural differences between the MtrR-ligand and MtrR-DNA complexes, the Fig. 3 should include three additional panels, each showing the indicated differences.

We thank the reviewer for this observation and have added a figure (Figure S5) showing the induction movements of MtrR. We have also included a supplementary morphing video of the MtrR induction movements.

3) Line 229: "the loop formed by residues 114-122 allows" – between which helices? Added to line: "the loop formed by residues 114-122, between helices $\alpha 6$ and $\alpha 7$, allows,..." (Line 225)

4) Table 1: the statistics of the Ramachandran plot should be included.

The Ramachandran statistics have been added to Table 1 (Page 29). Additionally, these statistics are included in the Methods section. (Lines 419-423)

Reviewer #2 (Remarks to the Author):

The manuscript by Hooks et al. reports the structures of MtrR, the transcriptional regulator of the MtrCDE efflux pump as well as several other genes/proteins in *N. gonorrhoeae*, bound to several steroid hormones, and interrogates the structural implications of hormone binding. The data show that MtrR directly binds progesterone, estradiol, and testosterone (and ethinyl estradiol found in some birth

control pills) and “activates” MtrR (i.e. causes release from its DNA binding sites), and that this activation can increase antibiotic resistance to other antibiotics or host antimicrobial peptides.

Overall, I found the evidence to be very convincing, the various approaches used to be appropriate and complementary, and believe the study illuminates fundamentally important aspects of gonococcal host-pathogen interactions. That said, there were areas where the presentation could be improved. Below are many comments (most are minor) from a careful reading that would help clarify the presentation.

1. The stoichiometries of binding for progesterone and estradiol in ITC experiments (and comparison to structural data) are confusing. For example, the stoichiometry of progesterone is reported as 0.5, but at least one of the 2 dimers in the asymmetric unit of the crystal structure shows each monomer with a bound progesterone (the other one has CAPS bound in one monomer, presumably from the crystallization solution) (Fig 2 and lines 176-177). Why is it different than the ITC experiments? The explanation provided does not address these contradictory data satisfactorily.

The inconsistency of PTR stoichiometry is addressed in lines 172-177: “This could be in part because MtrR is crystallized in an excessive concentration of steroids which does not reflect its more physiological binding, the apparent progesterone concentration in the ITC experiments is not correct, or that we are observing partial occupancy of each binding site in the crystal structure. Regardless, the MtrR-PTR complex assumes an induced conformation.”

The apparent PTR concentrations may be skewed due to its lower solubility or purity of the purchased molecule.

2. In Fig. 2, the estradiol binding experiments seems to show a molar ratio of <0.5, despite the $N \sim 1$ next to the graph. In fact, the highest molar ratio shown on the X axis is 0.5

Thank you so much for bringing this to our attention. The incorrect figure was included; the manuscript provides the correct values. The figure has been updated in the revision. (Page 23)

3. In Fig. 4, the 3 mutants appear to have a stoichiometry of binding for progesterone closer to 1 than 0.5.

The measured stoichiometry for MtrR mutants is: 0.81 (R176E) and 0.46 (D171A). But binding of these mutants to progesterone is significantly affected. Experimentally, we cannot reach saturation of the steroids binding to the mutants, thus the stoichiometry is not conclusive. (Fig 4)

4. It took me a while to understand the structural changes that occur in MtrR upon binding of steroid hormones (Fig 2B), in part because it looks like the HTH in the subunit on the left is pretty well aligned with that of the apo structure whereas the right side subunit is not. I was expecting to see the HTH motifs of the two subunits to be equally displaced from the apo structure. Is this a feature of the overlay procedure? Are the conformational changes in the two subunits identical? If they are, perhaps a more balanced overlay (in which the conformational changes are shown equally for the two subunits) would help make this clearer.

One protomer was superimposed (rather than an overlay of the entire dimer) to highlight the large movement within the dimer when comparing the two structures. But the conformational change is identical for each subunit. To address this, we have included a supplemental figure and movie to show the induction movements (Fig S5).

5. Line 136, the FP experiments of testosterone- and estradiol-bound MtrR show a 3-fold decrease in binding affinity for the mtrCDE operator (progesterone causes even less of a decrease, 2.3). These fold-decreases in affinity seem very small. Presumably it is sufficient to allow induction of transcription, but

perhaps the authors could comment and explain why such a small shift in affinity is physiologically relevant. Could this binding be influenced by the presence of RNA polymerase in the full promoter DNA? The reviewer raises an interesting point. We do not know the effects of RNA polymerase on binding and have not expressed or purified the *N.g.* RNAP to carry out such experiments. However, at higher concentrations of protein, MtrR begins to bind nonspecifically to DNA, possibly explaining why MtrR shows only a modest fold change in binding affinity for the *mtrCDE* operator compared to *rpoH*, since their affinities differ, i.e., the *rpoH* operator is bound tightly in the absence of steroid, whilst the binding to the *mtrCDE* operator is tight but approximately 6-fold weaker in the absence of these steroid inducers. Moreover, the addition of 1% methanol has a stronger effect on the binding of MtrR to the *mtrCDE* operator for reasons we do not understand. Thus, there are limitations experimentally as we cannot use higher concentrations of MtrR in the measurements. Nevertheless, there is a fold increase that is mirrored by cellular data and these steroids do indeed lead to induction.

6. Lines 201-203 discuss residues (K167, R176', and R137) that coordinate a phosphate group in MtrR. It seems to me that if they are interesting enough to mention, that they be shown in Fig 3C. Also, does phosphate (or similar ion) have a role in MtrR function, e.g. in stabilizing the MtrR dimer, or is it simply there from crystallization buffer, which is high in phosphate?

Phosphate is a main component of the crystallization buffer, so it is indeed present at high concentrations (approximately 1.6 M). We mention this binding because it is observed in our structures and proximal to the entrance of the ligand binding pocket. However, it is important to point out that our isothermal titration calorimetry binding experiments carried out in the presence of phosphate (data not shown), show no significant difference from experiments without phosphate, indicating the binding of a phosphate ion at this location is physiologically relevant.

7. Lines 251-252 has poor sentence construction: "which is further buttressed by the finding that neither cholesterol nor cortisol, which do not have a polar atom in a similar position, do not bind to MtrR (Fig S1). Neither x nor y...do not bind

Thank you. We have corrected this sentence.

8. Lines 270-272: "Deletion of *mtrD* encoding the inner membrane component of the efflux pump significantly lowered the MIC for PTR, EST, and TES, which indicated that these gonadal hormones are substrates of the MtrCDE pump (Table S1)." I would suggest removing "significantly", since no statistics are shown and except for Testosterone, the fold change is either 2-fold (considered not significant) or unknown (estradiol).

Thank you. We have edited this sentence as suggested.

9. The authors report no binding (NB) of progesterone to the W136A mutant (Fig. 4) but do report a binding affinity to the W136L mutant (Fig S5). However, when I look at those ITC graphs, the W136A mutant data (e.g. point to point variation) looks similar if even better in W136A than W136L, and also has a curve fit-why no Kd?

Binding experiments for MtrR W136L were not reproducible. Three replicates of the ITC experiments were completed with varying results and resulted in an unreliable curve-fits. Unlike MtrR W136A, which consistently showed no binding and could not be fit to a curve. Thus, thermodynamic data for these mutants is an average of poorly fit thermographs:

10. The authors should include WT binding data in the table in Fig S5.
 Edited to include WT Kd in the table of Fig S5.

11. While I agree that adding a sub-MIC level of steroid hormones decreases the capacity of Azi and Polymyxin B to inhibit GC growth, the magnitude of this effect is unclear to me. Do the MICs of the two antibiotics change? How do the dilution experiments translate to resistance?

MIC determination experiments can be done using either Agar dilution or broth dilution (or microdilution). In the case of our experiments presented in figure 5 the point of undiluted cultures (UN) is the same procedure that you would do in a GC-agar MIC determination experiment. If we observe the azithromycin result, we can say that the MIC for the WT in GC-agar is 0.06 and 0.125 when progesterone and estradiol are added. The *mtrR* mutant, however, remains unchangeable at 0.25 for the three conditions tested. That is how a technician would read the MIC in a clinical lab. Thus, we can say that the hormones increase the MIC for the WT 2-fold and therefore the resistance to azithromycin. These experiments were repeated two to three times with consistency. The plating of the dilutions is done to have a more visual representation of how much survival has the bacterial population as a whole in the assay. From that perspective we can conclude that the WT in the presence of the hormones survive the challenge of azithromycin significantly more (3 to 5 1/5 dilution points).

For polymyxin B the MIC of the WT in GC-agar would read at 100 μg/mL and that would change to 200 in the presence of the two concentrations of estradiol tested. The MIC of the *mtrR* mutant remained unchangeable at 200. Thus, we can conclude that the presence of estradiol increased the MIC of the WT to polymyxin by 2-fold, and using the dilution points we can say that in the presence of estradiol bacterial cells of the WT survive 25 to 125 times more (in relative CFU numbers) to polymyxin than in the absence of estradiol. From the experiment we can conclude that sublethal amounts of estradiol enhances bacterial resistance to polymyxin B.

12. Lines 296-297, "Importantly, genes not subjected to MtrR regulation (*lptA* and *ngo1249*) were not induced by these ligands (Fig S8)." Lines 462-463. "Steroidal hormones do not affect the expression of genes outside the MtrR regulon." Using 2 genes to imply that these hormones don't affect expression outside of the MtrR regulon seems an overreach. This really can't be answered without an RNAseq.

experiment. I'm not suggesting that one be done, just that the text be modified to reflect this limitation. We agree that the statement cannot be proven without an RNA-Seq experiment. Therefore, we have reworded the legend in Fig S9 to state: "Steroidal hormones did not affect the expression of two genes not regulated by MtrR, which validates that the differences we observed in expression levels for *mtrC* and *rpoH* in the presence of the hormones are due to MtrR regulation and not to a polar effect of the hormones at the locus where these recombinant genetic constructions were placed."

13. The data visualization in Fig 6 (as well as S7, S9, and S10), particularly the line graphs, nicely shows the effects of the mutations. These data are convincing that the D171A mutant does not respond to steroid hormones.

We thank the reviewer for the comment.

14. How specific is the D171A mutation for preventing hormone-promoted activation in MtrR? Do other non-steroidal inducers still activate MtrR?

The only previously studied ligands of MtrR are bile salts chenodeoxycholate (CDCA) and taurodeoxycholate (TDCA), found at extra-urogenital GC infection sites. Binding experiments with these compounds were not tested with MtrR D171A.

15. Lines 298-300, "In addition to the MtrR-D171A variant, similar results were obtained with FA19 strains producing the MtrR mutant proteins with single amino acid changes at positions 136 or 176 when grown in the presence of TES, EST or PTR (Fig S9)." This statement does not comport with the data shown. While I agree that the D171A mutant is not functional, the data with the other mutants are mixed. By my eye, I ascertained the following from the data in Fig 4, S7, S9, and S10:

D171A. No activation=>Estradiol, Progesterone, Testosterone, Ethinyl estradiol

W136A. Not done

W136L. Activation=> Estradiol, Testosterone (rpoH); Weak activation=> Testosterone (mtrC)

R176E. Activation=> Estradiol (rpoH); Weak activation=> Testosterone (rpoH), Progesterone (rpoH), Estradiol (mtrC); No activation=> Progesterone (mtrC)

The authors in fact don't really discuss the mutants sufficiently, although they show the data in the supplemental figures. This should be addressed.

We agree with the referee and have expanded on this discussion in the revision. (Lines 296 – 303).

Reviewer #3 (Remarks to the Author):

This manuscript describes analysis of the function of the MtrR regulatory protein. Using a number of in vitro assays, together with determining high resolution structures, insight has been provided into the identity of steroid compounds that can act as inducers of the system and which instigate a conformational switch in the regulator. The Mtr system is not just involved in the regulation of the *mtrCDE* operon but can act as a master regulator affecting multiple different factors both in a positive and negative fashion.

The authors have previously published work on MtrR, including structures bound to two regulatory sequences and a pseudo apo structure. In the latter, a detergent molecule was found in the binding site and therefore the protein was in the induced configuration. Identification of compounds that can induce the expression of a multidrug export protein can aid in discerning the function and biological importance of the export system, thus these findings are of interest.

It is intriguing to see different stoichiometries of drugs bound to MtrR. Three ligand bind with one ligand:protomer, while the ligand PTR appears to be 1:dimer. It is worded strongly that 1 PTR is unambiguously bound to one dimer however in the next sentences there is a caveat describing the possibilities that a CAPS molecule could be present or a second PTR. Another alternative that was posed is that a progesterone molecule may be in the asymmetric unit without explaining how this could occur. The wording could therefore be changed as these data are convincing as with the other ligands and the explanation for the interactions are not as clear cut.

Azithromycin, itself a substrate of the MtrCDE pump, doesn't not appear to bind to MtrR and therefore does not induce expression of the efflux system. An alternative hypothesis of the way the presence of the drug may still be recognised by the pump has been presented. It is probably beyond the scope of this manuscript, but it will be interesting in future to decipher this different mechanism.

We thank the reviewer for their comments and suggestions. Please note that CAPS is a biological buffer and not a detergent. We completely agree with the reviewer on the latter point about deciphering the azithromycin "induction" of this efflux pump. We hypothesize that one or more of the cysteines on MtrR is an ROS sensor that is able to respond to oxidation stress caused by antibiotic treatment and results in an "induced" conformation of MtrR, thus allowing transcription of the *mtrCDE* and *rpoH* genes. We think this is beyond the scope of this paper, but we are currently working on elucidating this role of MtrR, likely indirect via upregulating *rpoH*, thereby allowing GC a defence against the drug azithromycin.

Specific corrections.

Line 18 and 69 - Multiple transferable resistance Repressor should read multiple transferable resistance repressor **Changed as suggested.**

Line 34 - gram-negative should read Gram-negative **Done**

Line 112 - should read a significant decrease. **Done**

Line 160 -162 is a repetition of previous findings and could be deleted/made more succinct. **We prefer to keep this information as presented as specific residues are called out in other places in the text and this allows an easy guide for a reader to locate them in the context of each subunit. Furthermore, I would imagine that not every new reader will have read all of our previous work on MtrR.**

Line 245 - W137A should be W136A **Done**

Line 246 - W137L should be W136L **Done**

Line 295 - Using the nomenclature MtrR+ could signify overexpression of MtrR protein. If this is not what was intended, it would be better to change this to WT or to signify if it was a complemented strain. **Okay, we have changed this as suggested by the reviewer.**

Line 304 – Neisseria should be N. **Done**

Line 327 - th should be the **We thank the reviewer for this catching the typo.**

Line 355 - Staphylococcus should be S. **Done**

Line 356 - Escherichia should be E. **Done**

Line 513 – should it be and azithromycin? Yes, changed as suggested.

- 1 Lucas, C. E., Balthazar, J. T., Hagman, K. E. & Shafer, W. M. The MtrR repressor binds the DNA sequence between the mtrR and mtrC genes of *Neisseria gonorrhoeae*. *J Bacteriol* **179**, 4123-4128 (1997). <https://doi.org:10.1128/jb.179.13.4123-4128.1997>
- 2 Folster, J. P. *et al.* MtrR modulates rpoH expression and levels of antimicrobial resistance in *Neisseria gonorrhoeae*. *J Bacteriol* **191**, 287-297 (2009). <https://doi.org:10.1128/JB.01165-08>
- 3 Beggs, G. A. *et al.* Structures of *Neisseria gonorrhoeae* MtrR-operator complexes reveal molecular mechanisms of DNA recognition and antibiotic resistance-conferring clinical mutations. *Nucleic Acids Res* **49**, 4155-4170 (2021). <https://doi.org:10.1093/nar/gkab213>

Reviewer #1 (Remarks to the Author):

The authors addressed all questions and remarks. I recommend the publication of the article in its current form.

Reviewer #2 (Remarks to the Author):

The revised manuscript by Hooks et al. has done a very nice job of responding to the concerns of the reviews. The response letter is detailed and adequately deals with the issues raised in the initial review. I appreciate the video showing the structural changes following hormone binding, it really clarifies the mechanism of release from repression. After a careful reading of the revised text and figures, I found no concerns.

Reviewer #3 (Remarks to the Author):

I appreciate the changes made to address all of the reviewers' comments. I have not identified any further modifications required to be made.